# On the Recovery of Hematite from an Iron Ore Fine Fraction by Electroflotation Using a Biosurfactant

**Carolina R. Simões, Ronald R. Hacha, Antonio G. Merma and Maurício L. Torem ***

Department of Chemical Engineering and Materials, Pontifical Catholic University of Rio de Janeiro, Rio de Janeiro RJ 22453-900, Brazil; carolina.simoesca2@gmail.com (C.R.S.); rojas@esp.puc-rio.br (R.R.H.); anguz21@hotmail.com (A.G.M.)
*   Correspondence: torem@puc-rio.br

**Abstract:** Electroflotation is a clean technique potentially able to recover fine particles from mineral suspensions. The aim of the present work was to evaluate the electroflotation of fines and ultrafine particles of an itabiritic iron ore using a biosurfactant extracted from *Rhodococcus opacus* bacteria. Infrared spectroscopy and zeta potential measurements confirmed the interaction between the biosurfactant and the mineral surface. The isoelectric point of hematite presented a value of about pH 5.3; after interacting with the biosurfactant, a charge reversal point of pH 3.5 was observed. The biosurfactant reduced the air/water surface tension from 71 to 40 mN/m, using 25 mg/L concentration. The electroflotation process of fine and ultrafine particles was evaluated as a function of pH, biosurfactant concentration, stirring of the aqueous suspension and current density. It was observed that the iron recovery (%) and iron grade (%) were negatively affected by increasing pH value. Therefore, best results were achieved at pH 3. Biosurfactant concentration and current density positively affected both response variables. An iron recovery value of about 83% and an iron grade of about 59% were achieved for the −38 + 20 μm size fraction; whereas, higher values were attained (98% and 64%, respectively) for the finer size fraction −20 μm.

**Keywords:** electroflotation; iron ore; ultrafine particles; biosurfactant; *R. opacus*

## 1. Introduction

In mineral processing operations large quantities of waste are generated, and these residues are mainly constituted of sterile mineral and a huge fraction of ultrafine particles. The mineral of interest presented in the ultrafine fraction is poorly recovered by conventional flotation techniques [1]. Thus, this material that may have iron content above 50% redirected to tailings dams and may cause disruption, causing economic and environmental impacts [2]. It is believed that the poor recovery efficiency of ultrafine particles is due to the low probability of bubble–particle collision and interaction, high particle–bubble energetic barrier, high surface area and a high consumption of reagents [3–5]. In addition, ultrafine particles are expected to present a higher degree of entrainment and entrapment during flotation operations [6].

Bubble size distribution together with particle size are some of the main factors affecting flotation efficiency. This is because adhesion, collision and transport stages are directly related to both factors. Additionally, recovery of fine particles is more effective when bubbles with lower size are presented in the system, because of their higher surface area [7,8], suggesting that the flotation of fine particles can be mostly enhanced by improving the collision probability [9].

Several technologies have been used to increase the efficient recovery of ultrafine particles; one of them is electroflotation [10]. This electrochemical technique could be considered as a clean technology, where oxygen and hydrogen micro-bubbles are electrolytically produced. The Sauter diameter of these

micro-bubbles is between 17 and 105 μm [11]. According to Bagotsky [12], several redox reactions might be present in the solution and on the electrode surface during water electrolysis. However, a simpler approach was considered in the present work, the redox reactions for water electrolysis are presented in Equations (1)–(3). Oxygen bubbles are produced on the anode electrode during the oxidation reaction (Equation (1)), while hydrogen bubbles are produced on the cathode electrode during the reduction reaction (Equation (2)). The global electrolysis reaction is presented in Equation (3).

$$\text{Oxidation } 2OH^- \rightarrow H_2O + \frac{1}{2}O_{2(g)} + 2e^- \tag{1}$$

$$\text{Reduction } 2H_2O + 2e^- \rightarrow 2OH^- + H_{2(g)} \tag{2}$$

$$\text{Global reaction } H_2O \leftrightarrow H_{2(g)} + \frac{1}{2}O_{2(g)} \tag{3}$$

In electroflotation it is possible to control the production rate and bubble size of the produced gases by modifying the current density, pH of the system, the material and type of electrodes, and the electrolyte concentration [13,14]. Consequently, electroflotation is a thought-provoking flotation alternative to process fines and ultrafine mineral particles. It is also possible to turn it into an eco-friendly approach, by using bioreagents (microorganisms and their by-products) as collectors, modifiers and frothers.

Biosurfactants may present comparable characteristics as synthetic surfactants used in mineral flotation. The adsorption of collectors at the solid–liquid interface imparts and enhances a mineral's hydrophobicity [15]. This means that biosurfactants have amphipathic character, are able to adsorb onto mineral surfaces, modifying their surface properties, and consequently changing their hydrophobic degree. It is believed that biosurfactants are biodegradable and compatible with the environment, present low toxicity and can be produced using renewable substrates [16]. According to Mesquita [17] the hydrophobic degree of microorganisms and their by-products is attributed to their high fatty acid and proteins content, and to the cell-wall acid/basic character, whilst the hydrophilic degree is related to their polysaccharides content. Thus, the present work aims to study the electroflotation of fine and ultrafine particle of iron ore using the biosurfactant extracted from *R. opacus* bacteria. It is demonstrated to be a clean and important alternative in the recovery of slimes, which are currently deposited in tailing dams.

## 2. Materials and Methods

### 2.1. Mineral Samples

In this study, two kinds of sample were used: pure mineral samples (hematite and quartz) and fine particles from the itabiritic ore for the flotation tests. Those samples were, respectively, supplied by a mining plant located at the so-called "Iron Quadrangle", Minas Gerais. Hematite and quartz (−20 μm) were used for the fundamental studies as zeta potential and infrared analysis.

The fine particle samples from the itabiritic ore were obtained from the desliming processes (overflow) and classified in two particle-size fractions: Fraction A: −38 + 20 μm and fraction B: −20 μm. These samples were characterized by chemical (titration with potassium dichromate) and mineralogical (X-ray diffraction, Bruker-AXS) analysis.

### 2.2. Preparing and Obtaining the Biosurfactant

The *Rhodococcus opacus* bacteria was provided by the Brazilian Collection of Environmental and Industrial Microorganisms (CBMAI) supplied by the Chemical, Biological and Agricultural Pluridisciplinary Research Center (CPQBA-UNICAMP). The bacterium was grown in a YMG (Yeast Malt Glucose) medium, composed of glucose (20 g/L), peptone (5 g/L), malt extract (3 g/L), and yeast extract (3 g/L), using a rotary shaker at 150 rpm for 7 days. After they were grown, the cells were separated

from the broth by centrifugation at 4500 rpm for 10 min. The biomass obtained was washed twice with deionized water, then re-suspended in ethylic alcohol (98%) solution, and stored overnight in a refrigerator. Then, the suspension was autoclaved for 20 min. at 1 atm. After that, the solution fraction (biosurfactant-rich solvent) was separated (centrifuged and filtered) from solid fraction and was dried at 50 °C. Finally, the latter was dissolved in deionized water, and it was stored in the fridge.

### 2.3. Surface Properties of the Mineral

The surface tension measurements of the air/solution interface were carried out using a tensiometer DC200A Surface Optics by the Nöuy ring method. The parameters studied were pH (3 to 11) of solution and biosurfactant concentration (25 to 1000 mg/L).

The zeta potential measurements of pure minerals before and after interacting with biosurfactant were carried out using the electrophoretic zeta-meter system +4.0. The suspension pH was evaluated from 3 to 11, aliquots of HCl (0.1 mol/L) and NaOH (0.1 mol/L) were used to adjust the pH and NaCl was used as indifferent electrolyte. The conditioning of the mineral was undertaken using 300 mg/L of the biosurfactant for 5 min, then the suspension was washed with deionized water, and finally, the material was filtered and dried at 50 °C.

Infrared analyses were performed on a Scientific Nicolet 6700 Fourier transform infrared (FTIR) spectrophotometer using the KBr pellet method. Pure minerals (hematite and quartz) were analysed before and after interacting with biosurfactant. The mineral to KBr ratio was 1/200 (wt./wt.). The spectra were performed at a resolution of 4 cm$^{-1}$ using 360 scans.

### 2.4. Electroflotation Test

For these tests, a modified *Partridge-Smith* binary cell (Figure 1) was used. The dimensions of each glass half-cell were of 50 mm external diameter, 46 mm internal diameter and 215 mm height. A Millipore mesh of 40 mm in diameter was added in each one and the union (saline bridge) was made with a 44 mm diameter rubber. The union between the semi-cells was made with a rubber sheet of 44 mm. The configuration of the cell made it possible to use oxygen bubbles or hydrogen bubbles; for the present study, only the hydrogen bubbles were considered.

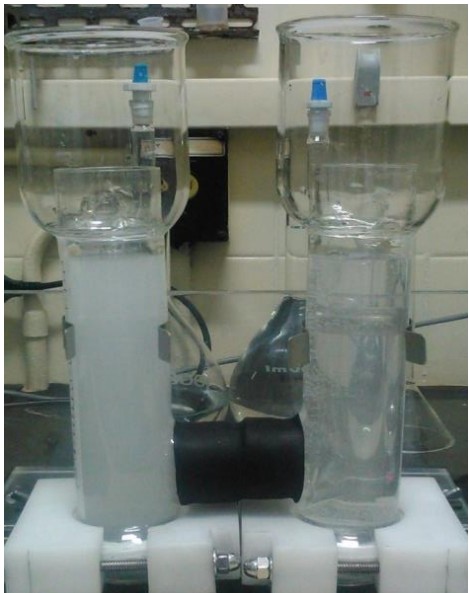

**Figure 1.** Modified *Partridge-Smith* binary cell.

The electrodes were configured as platinum mesh (99.5% purity), presenting a surface area of 25 cm$^2$, a wire diameter of 0.08 mm, and a gap between electrodes of 0.16 mm (Figure 2).

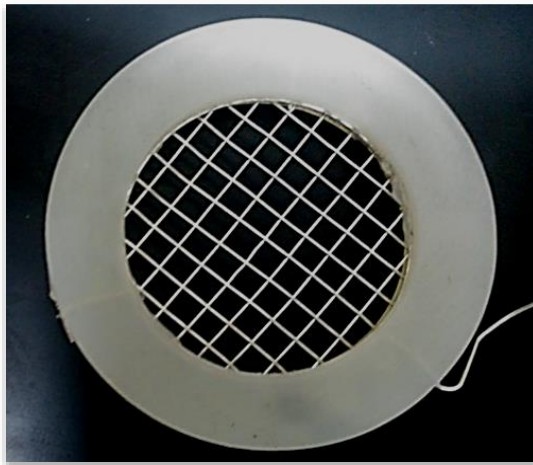

**Figure 2.** Platinum electrode.

Each semi-cell was filled with electrolytic solution (0.2 mol/L Na$_2$SO$_4$), and then electrolysis was initiated in order to generate the hydrogen bubbles. Then, the experimental conditions (pH, current density) were adjusted according to the desired values. In parallel, conditioning of the sample ore (1 g) with the biosurfactant was carried out for 5 min. After that, the conditioned mineral was placed inside the cathodic semi-cell, electrolytic solution was added until the useful volume was achieved, and then, the electroflotation test was carried out for 10 min. The parameters studied were pH (3 to 11), current density (5.28 to 16 mA/cm$^2$), biosurfactant concentration (50 to 800 mg/L), agitation (300 to 700 rpm), and particle size fraction (A, B).

## 3. Results

### 3.1. Iron Ore Characterization

Table 1 presents the chemical analysis of the two iron ore fractions. The results indicated that the content of Fe in the samples was 53.84% and 54.06%, for fraction A and B, respectively. The correspond iron values are presented in Table 1. The chemical analysis by titration with potassium dichromate presented similar values, which demonstrates the correlation between both techniques. It was observed that the finer fraction presented higher iron content, which is in accordance with literature.

**Table 1.** Chemical analyses of iron ore sample.

| Sample | Composition | | |
|---|---|---|---|
| | Fe (%) | Fe$_2$O$_3$ (%) | SiO$_2$ (%) |
| A: −38 + 20 μm | 53.84 ± 3 | 76.91 | 23.09 |
| B: −20 μm | 54.06 ± 3 | 77.22 | 22.78 |

The X-ray diffraction spectra revealed that the main mineralogical phases are hematite (F$_2$O$_3$) and quartz (SiO$_2$); the spectra can be found elsewhere [18].

### 3.2. Fourier Transform Infrared (FTIR) Spectroscopy

Figure 3 shows the infrared spectra of the hematite before and after interacting with the biosurfactant. Before the interaction, it was possible to observe bands corresponding to the hydroxyl groups (-OH) and Fe-O stretching vibrations. The presence of peaks in this region (3467.26 cm$^{-1}$) could be related to hydrogen bonds due to the presence of residual water on the sample surface [19]. The peaks at 589.22 cm$^{-1}$ and 466.94 cm$^{-1}$ can be attributed to the vibrations of the Fe-O bond, characteristic of hematite. Some authors pointed out that the bands below 1000 cm$^{-1}$ are characteristic

of iron oxides [20,21]. After the interaction, it was observed that the characteristic bands corresponding to the stretching mode of Fe-O bonds of hematite were maintained (589.22 cm$^{-1}$ and 466.94 cm$^{-1}$). On the other hand, modifications in the hematite spectrum could confirm the hydrophobization of the hematite surface. The band at 3451.64 cm$^{-1}$ indicates the vibration of the stretch of groups -OH of polysaccharides and proteins [22]. The peak at 3017.52 cm$^{-1}$ corresponds to the asymmetric stretch of CH$_2$ and the band at 1632.92 cm$^{-1}$ may be related to the overlap of the C=O stretch found in lipids and triglycerides [17]. The band at 1403.76 cm$^{-1}$ may correspond to symmetric vibrations of the COO- group and at 1316 cm$^{-1}$ may correspond to asymmetric vibrations of the phosphodioxy group (PO$_2$$^-$) found in phospholipids and nucleic acids [23]. Similar approaches have been discussed in previous works [11,24]. This could confirm the results obtained in the present analysis.

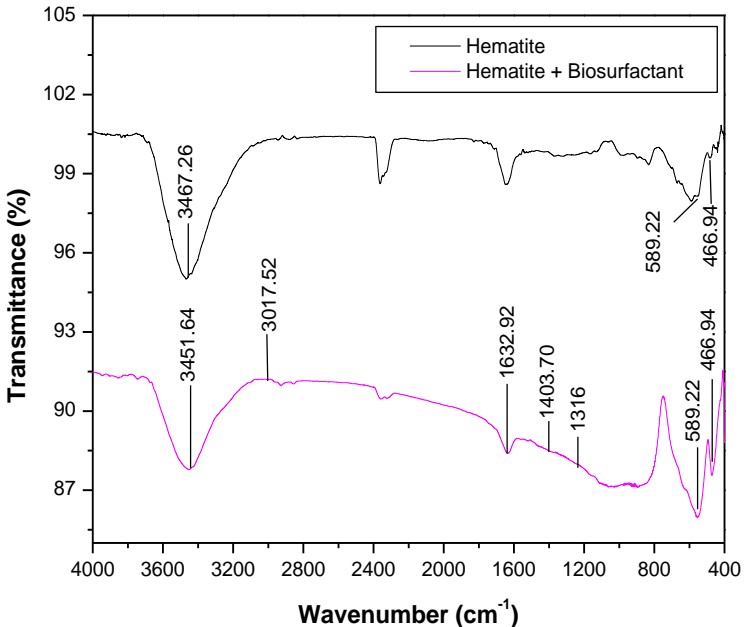

**Figure 3.** Fourier transform infrared (FTIR) spectra of hematite before and after interaction with biosurfactant.

Figure 4 shows the FTIR spectra of quartz before and after interacting with the biosurfactant. The quartz surface presented characteristic bands of the -OH group (3468.73 cm$^{-1}$) attributed to hydrogen bonds from OH or formed residual water [19,25]. The peaks between 1200–400 cm$^{-1}$ are characteristic of quartz (Si-O bonds) [25,26]. After interacting, it is possible to observe no change in the spectra, especially in the region of characteristic bands of quartz. This could be related with low or no adsorption of the biosurfactant onto the quartz surface. Merma [24] and Deo [27] pointed out a similar behaviour; the authors studied the adsorption of microorganisms onto hematite and quartz surfaces. Deo [27] identified the presence of the C-OH group, characteristic of polysaccharide, on the hematite surface, after interacting of the mineral with *Paenibacillus polymyxa* bacteria whilst no adsorption was identified on the quartz surface.

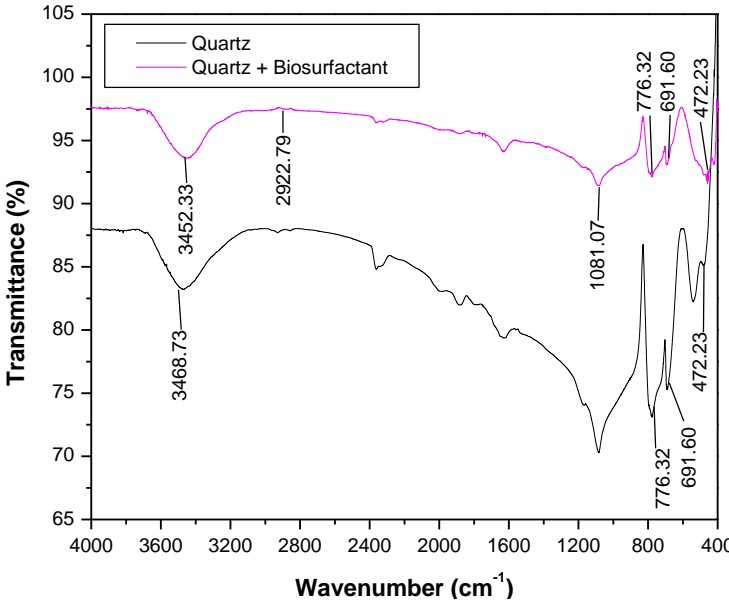

**Figure 4.** FTIR spectra of quartz before and after interaction with biosurfactant.

### 3.3. Surface Tension Studies

Figure 5 presents the effect of biosurfactant concentration on surface tension of the air/water interface, at pH 3. The presence of biosurfactant decreased the surface tension from 71.0 to 28 mN/m, the CMC (Critical Micelle Concentration) was identified at 200 mg/L, but the higher drop (~40 mN/m) was observed using a lower biosurfactant concentration (25 mg/L). The reduction in surface tension could be attributed to the amphipathic character of the biosurfactant extracted from the *R. opacus*. According to Szymanska and Sadowski [28], biosurfactants have an amphipathic nature capable of being easily absorbed at the air/water interface, reducing its surface tension and favouring the formation of foams by becoming biofoam.

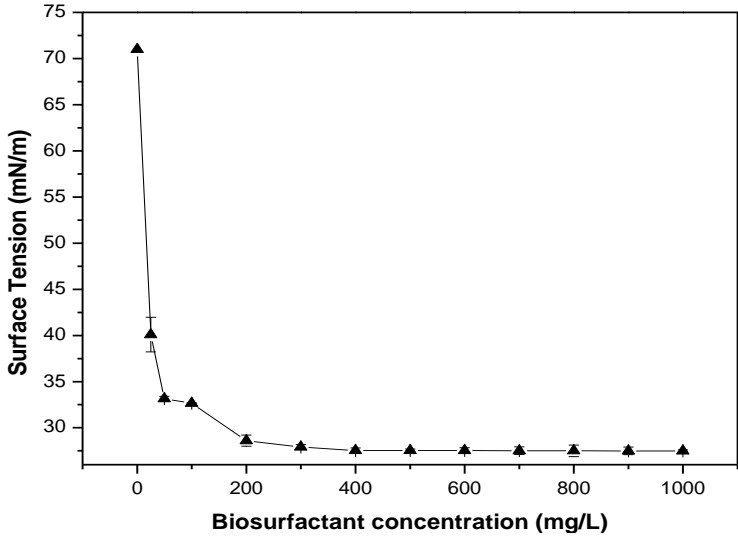

**Figure 5.** Influence of biosurfactant concentration on surface tension at pH 3.

Figure 6 shows the effect of pH on surface tension using a biosurfactant concentration of 300 mg/L. According to previous works [24,26] the surface tension is significantly affected by the pH of the system, especially when microorganisms were used as a collector. At pH around 11, the presence of microorganisms slightly affected the surface tension. A different behaviour was observed with the

biosurfactant, and the presence of biosurfactant greatly affected the surface tension of the solution at the whole pH range (Figure 5). The higher effect was observed at around its isoelectric point (IEP), pH 3, achieving a value around 28 mN/m, while at pH 11 a value of 37 mN/m was observed. This phenomenon suggests that at acidic pH, greater adsorption of the biosurfactant occurs at the air/water interface, reducing surface tension and, therefore, favouring the formation of foam. According to Makri and Doxastakis [29], the pH is the determining factor for foam stability, the best pH value for the formation of stable foam is around its IEP, which is also observed in the present work.

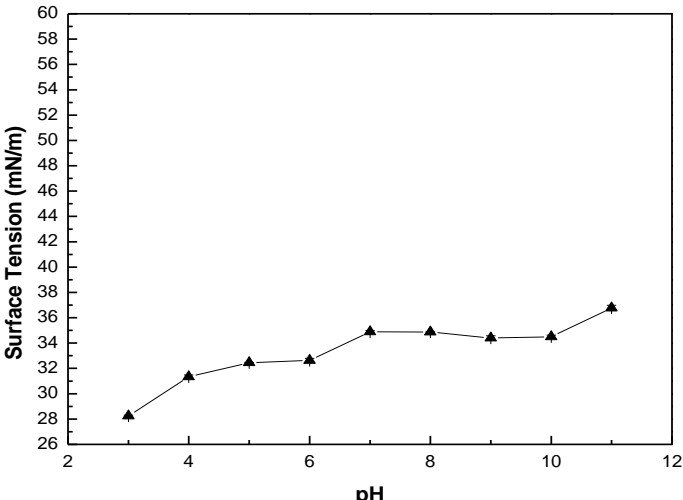

**Figure 6.** Effect of pH on surface tension of biosurfactant solution. Biosurfactant: 300 mg/L.

### 3.4. Zeta Potential Studies

Figure 7 presents the zeta potential of hematite before and after interacting with the biosurfactant. The hematite presented an IEP around pH 5.3 [18], and below the IEP the hematite presented a positively charged surface, achieving 10 mV at pH 3. Above the IEP a negatively charged surface was observed, achieving −27 mV at pH 11. Thus, this flexibility of hematite makes it possible to interact with anionic and cationic surfactants. It is believed that this biosurfactant have a complex structure and can be composed of mono-, di-, tri-, tetramicolates, making them present anionic characteristics [20]. Therefore, attractive electrostatic interactions would be possible below the hematite IEP [30,31]. The BS–hematite interaction modified the zeta potential profile of hematite, and more negative values were observed. In addition, a reversal charge point (RCP) was observed at about pH 3.5. According to Lang and Philp [32], these effects can be justified by the adsorption of the surfactant onto the mineral surface.

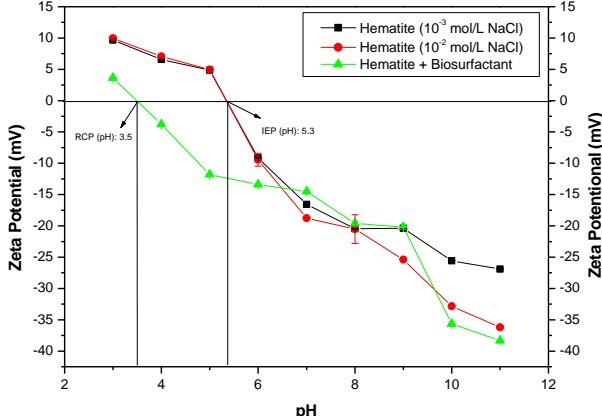

**Figure 7.** Zeta potential profiles of hematite before and after interacting with biosurfactant (300 mg/L).

Figure 8 shows the zeta potential profile of quartz, before and after interacting with the BS (biosurfactant). The quartz IEP was observed elsewhere [18], presenting a value of about pH 2.1. After the interaction, the profile presented more negative values and a RCP at pH 1.3. Once again, these behaviours could be related to the adsorption of the BS onto the quartz surface [33]. Olivera [26] and Camarate [34] evaluated the zeta potential of quartz after interaction with biosurfactants extracted from *R. erythropolis* and *Candida stellata*, respectively. Both of them found a reversal charge point after the interaction, at about pH 1.9 and about 2.15, respectively. The authors justified that these RCPs are due to attractive electrostatic interaction between the positively charged surface of the mineral and anionic groups presented on the biosurfactant.

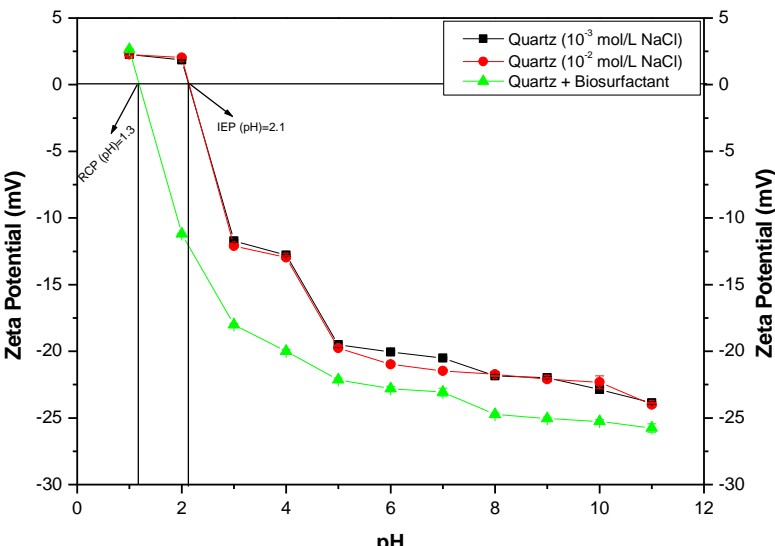

**Figure 8.** Zeta potential profile of quartz, before and after interaction with biosurfactant (300 mg/L).

## 3.5. Electroflotation Test

### 3.5.1. Effect of pH

Figure 9 shows the effect of pH on the electroflotation of iron ore. Two particle size fractions were studied (Fraction A: −38 + 20 μm, and fraction B: −20 μm) in Figure 9A,B, respectively. It can be observed that the highest iron recovery and grade were attained at pH 3; increasing pH negatively affected both response variables until pH 7, beyond that value the responses started to increase again, until pH 11. The iron recovery achieved values of around 80% and 98% for fractions A and B, respectively, while the iron grade presented values of about 59% and 64% for fractions A and B. This behaviour is in accordance with the zeta potential results, where the higher electrostatic interaction was around pH 3, close to the biosurfactant IEP. According to Kim [35], most of the substances present in the biosurfactant (polysaccharides, fatty acids, phospholipids and amino acids) can be activated at pH 3. This may contribute to the adsorption of biosurfactant on the mineral surface, favouring the floatability of particles. It is believed that several mechanisms are involved during the biosurfactant adsorption onto the mineral surface, which could be dependent on the pH of the system [11,24].

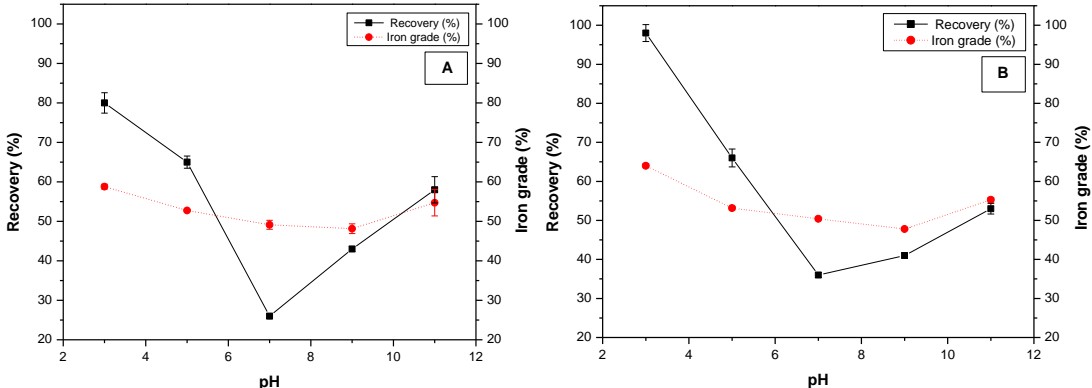

**Figure 9.** Electroflotation of iron ore as a function of pH. Current density 16 mA/cm$^2$, Biosurfactant: 300 mg/L; agitation: 300 rpm. (**A**): Particle size −38 + 20 μm. (**B**): Particle size −20 μm.

### 3.5.2. Effect of the Biosurfactant Concentration

The effect of the biosurfactant concentration on iron recovery and grade, during the electroflotation of iron ore fractions A and B, is shown in Figure 10. The increase in biosurfactant concentration positively affected the iron recovery and iron grade up to the concentration of 300 mg/L. The iron recovery achieved values of about 80% and 93% for fractions A and B, respectively, whereas the highest iron content was 59% and 64% for fractions A and B, respectively. Beyond that concentration, the effect was contrary, which may be related to the CMC. Thus, micelles or aggregates started to form prejudicing flotation efficiency, due to a reduction of the effective adsorption area. In addition, the biosurfactant could adsorb at the gas/liquid interface, reducing its interaction onto mineral surface [11,24,36].

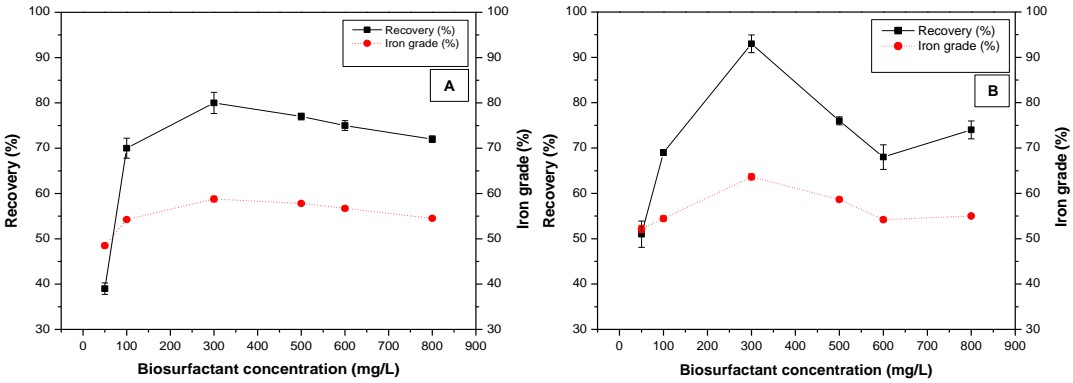

**Figure 10.** Electroflotation of iron ore as a function of biosurfactant concentration. Current density 16 mA/cm$^2$, pH 3, agitation 300 rpm. (**A**): Particle size: −38 + 20 μm. (**B**): Particle size: −20 μm.

### 3.5.3. Effect of Current Density

Figure 11 shows the electroflotation of iron ore (fractions A and B) as a function of current density. According to these diagrams, the higher the current density the higher the iron recovery and iron grade. This effect was appreciated for both iron ore fractions. However, higher values of iron recovery (93%) and grade (65%) were obtained for fraction A, using 16 mA/cm$^2$. It is believed that the current density is one of the main factors affecting the electroflotation efficiency. By modifying the current density, it is possible to control the redox reactions of an electrolytic cell, consequently making it possible to control the gas production rate, bubble nucleation rate, number of bubbles and diameter of the bubbles' gas. Thus, the higher the current density the higher the number of bubbles and the smaller the bubbles' diameter. This would incite a higher probability of collision between bubbles and particles and, therefore a higher probability of attachment; consequently, the electroflotation efficiency would increase [13,37,38].

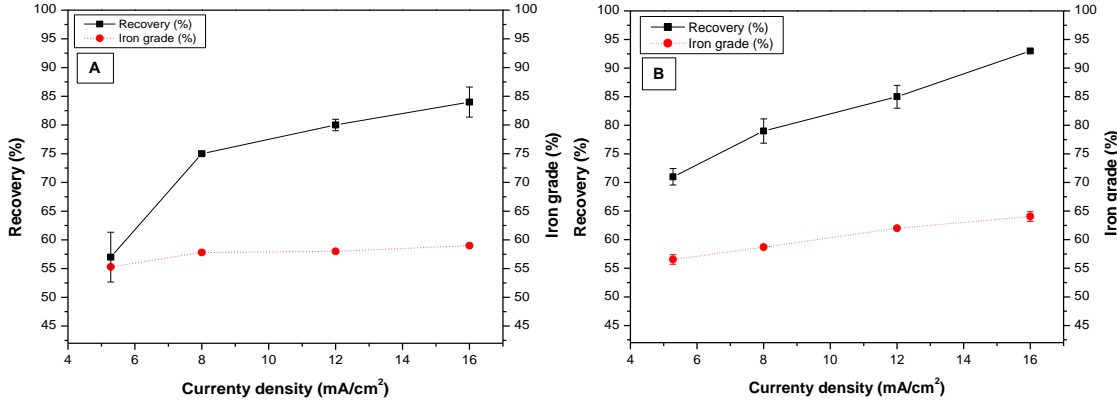

**Figure 11.** Electroflotation of iron ore as a function of current density. Biosurfactant: 300 mg/L, pH 3, agitation 300 rpm. (**A**): Particle size −38 + 20 μm. (**B**): Particle size −20 μm.

### 3.5.4. Influence of System Agitation

Figure 12 presents the effect of system agitation on the electroflotation of the two iron ore fractions. It is observed that the higher the agitation of the system the lower the electroflotation efficiency. This effect is more remarkable on the finer fraction (fraction B). As shown previously the highest iron recovery and grade were obtained for the particle size fraction B, with 300 rpm, achieving values of 93% and 64%, respectively. Lower values were attained with the fraction A (83% and 58%, respectively). These results are in accordance with literature, where it is known that excessive agitation levels generate the rupture of the bubble–particle aggregate, prejudicing their transport and recovery. Modirshahla [39] claimed that agitation is important for the bubble–particle collision and aids the transport phenomenon. Nevertheless, with high agitation levels the aggregates formed may collide and break up, which may provoke a drop in the flotation efficiency.

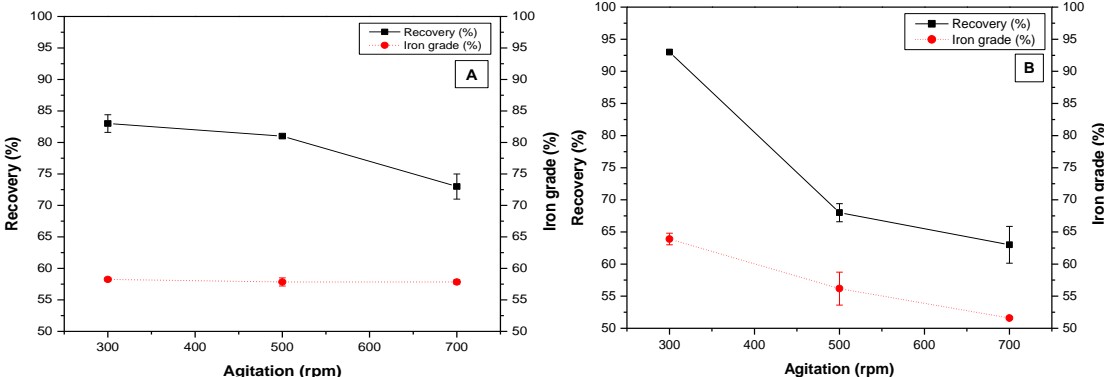

**Figure 12.** Effect of system agitation on iron ore electroflotation. Current density 16 mA/cm$^2$, Biosurfactant: 300 mg/L, pH 3. (**A**): Particle size −38 + 20 μm. (**B**): Particle size −20 μm.

## 4. Conclusions

The present results indicated the feasibility of the electroflotation process for the recovery of fine and ultrafine iron ore particles using the biosurfactant extracted from *Rhodoccocus opacus*. The FTIR spectra showed characteristic functional groups of the biosurfactant in hematite after interacting, such as NH, $CH_2$, C=O, COO-, $PO_2^-$. The zeta potential measurements showed a conceivable electrostatic interaction between hematite surface and biosurfactant, and a reversal charged point was identified at pH 3.5.

The uppermost electroflotation efficiency was attained, in this work, at pH around 3 and using 300 mg/L, achieving an iron recovery and iron grade of about 98% and 64%, respectively. Current density was the main factor affecting electrocoagulation efficiency. The principal reason was the control

of the rate bubble production, with smaller diameter. The electroflotation efficiency of the −38 + 20 µm fraction was lower than the −20 µm fraction. It was possible to achieve an iron recovery of about 98% and an iron grade of about 64%.

**Author Contributions:** Conceptualization, C.R.S., R.R.H. and M.L.T.; Data curation, R.R.H.; Funding acquisition, M.L.T.; Investigation, C.R.S.; Methodology, C.R.S., R.R.H. and A.G.M.; Resources, M.L.T.; Supervision, M.L.T.; Validation, R.R.H.; Writing—original draft, C.R.S. and A.G.M.; Writing—review and editing, C.R.S., A.G.M. and M.L.T. All authors have read and agreed to the published version of the manuscript.

**Funding:** This research was funded by Rio de Janeiro State Research Foundation, grant number E-26/202/811/2017 and National Council for Scientific and Technological Development, grant number 304639/2016-8 and 422450/2016-2.

**Acknowledgments:** The authors acknowledge Pontifical Catholic University of Rio de Janeiro, CNPq (National Council for Scientific and Technological Development); CAPES (Coordination for the Improvement of Higher-Level Personnel) and FAPERJ (Rio de Janeiro State Research Foundation) for economic and technological support.

**Conflicts of Interest:** The authors declare no conflict of interest. The funders had no role in the design of the study; in the collection, analyses, or interpretation of data; in the writing of the manuscript: or in the decision to publish the results.

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
