# Peer review of "On the Recovery of Hematite from an Iron Ore Fine Fraction by Electroflotation Using a Biosurfactant"

_minerals, doi:10.3390/min10121057_

Round 1

Reviewer 1 Report

The paper “On the recovery of hematite from an iron ore fine fraction by electroflotation using a biosurfactant” investigates electroflotation of fines and ultrafines particles of an itabiritic iron ore using a biosurfactant extracted from Rhodococcus opacus bacteria.

The paper presents enough quality to be published in the “Minerals, special issue Mineral Processing and Metallurgy”.

However, I suggest the following modifications:

  1. Units should be separated by the numbers by a space, i.e., line 119: 0.2mol/L – correct: 0.2 mol/L. All text should be revised.
  2. Line 126: 16mA/cm2 – correct: 16 mA/cm2.
  3. It is missing several Spaces between words, i.e., line 144: 466.94 cm1can. All text should be revised.
  4. Line 172. “Error! Reference source not found” should be removed and the sentence revised.
  5. Line 266. Figure 11 and not Figure 1.
  6. Line 280. Figure 12 and not Figure 2.

Author Response

Dear Editor,

Enclosed please find the attachement file for the reviewer 1.

Reviewer 2 Report

The authors Simões et al. have investigated the electroflotation of iron ore fines using a biosurfactant extracted from Rhodococcus opacus bacteria. The interactions between the biosurfactant and mineral surface is confirmed by the IR spectroscopy and zeta potential measurements. The present study is valuable as the resources are fast depleting and the low-grade ores are needed to concentrate on their efficient exploitation. In this context, the experimental works in the present manuscript are well-conducted and expressed, however, there are many typos that certainly be rectified by the authors. Except that, the authors have some suggestions as below:

  1. The authors must include the PZC of iron ore fines and yielded hematite concentrate.
  2. Add isoionic point (IIP) of the biosurfactant.
  3. Besides other typos, mainly related to space between two words, the typo in line 172 needs attention.
  4. The reviewer would like to suggest to modify the tile of the manuscript as technically there is no recovery, so better to avoid “recovery” in the title.
  5. The authors may cite some useful references in the introduction, as suggested: https://doi.org/10.1016/j.mineng.2020.106277; https://doi.org/10.1016/j.mineng.2020.106368

Author Response

Dear Editor, 

Enclosed please find the attached file for reviewer 2. 
